# The Structural Characteristics of Compounds Interacting with the Amantadine-Sensitive Drug Transport System at the Inner Blood–Retinal Barrier

**DOI:** 10.3390/ph16030435

**Published:** 2023-03-13

**Authors:** Yusuke Shinozaki, Yuma Tega, Shin-ichi Akanuma, Ken-ichi Hosoya

**Affiliations:** Department of Pharmaceutics, Graduate School of Medicine and Pharmaceutical Sciences, University of Toyama, 2630 Sugitani, Toyama 930-0194, Japan; d2262304@ems.u-toyama.ac.jp (Y.S.); tega@pha.u-toyama.ac.jp (Y.T.); akanumas@pha.u-toyama.ac.jp (S.-i.A.)

**Keywords:** blood–retinal barrier, inner BRB, amantadine, retinal diseases, drug transport system

## Abstract

Blood-to-retina transport across the inner blood–retinal barrier (BRB) is a key determinant of retinal drug concentration and pharmacological effect. Recently, we reported on the amantadine-sensitive drug transport system, which is different from well-characterized transporters, at the inner BRB. Since amantadine and its derivatives exhibit neuroprotective effects, it is expected that a detailed understanding of this transport system would lead to the efficient retinal delivery of these potential neuroprotective agents for the treatment of retinal diseases. The objective of this study was to characterize the structural features of compounds for the amantadine-sensitive transport system. Inhibition analysis conducted on a rat inner BRB model cell line indicated that the transport system strongly interacts with lipophilic amines, especially primary amines. In addition, lipophilic primary amines that have polar groups, such as hydroxy and carboxy groups, did not inhibit the amantadine transport system. Furthermore, certain types of primary amines with an adamantane skeleton or linear alkyl chain exhibited a competitive inhibition of amantadine uptake, suggesting that these compounds are potential substrates for the amantadine-sensitive drug transport system at the inner BRB. These results are helpful for producing the appropriate drug design to improve the blood-to-retina delivery of neuroprotective drugs.

## 1. Introduction

The inner blood–retinal barrier (BRB), which consists of retinal capillary endothelial cells, directly separates the retina from the blood. Although these cells form tight junctions with each other and restrict the paracellular transport of compounds at the inner BRB, various transport systems are known to exist to facilitate nutrient and drug distribution [1,2]. For example, glucose transporter 1 (GLUT1/slc2a1), L-type amino acid transporter 1 (LAT1/slc7a5), creatine transporter (CRT/slc6a8), and riboflavin transporters 2-3 (RFVT2-3/slc52a2-3) are expressed at the inner BRB and contribute to the retinal transfer of drugs/nutrients from the blood [1,2,3]. In addition, several cationic drugs, such as propranolol, verapamil, and clonidine, have been reported in the literature to be actively distributed to the retina across the inner BRB by putative cationic drug transport systems [3,4]. Carrier-mediated transport via the inner BRB is a key determinant of drug concentration and pharmacological effect in the retina; thus, a better understanding of drug transport systems at the inner BRB would be helpful in developing appropriate strategies for efficient retinal drug delivery.

At the inner BRB, we recently reported that a carrier-mediated transport system contributes to the retinal transport of the cationic compound amantadine, an adamantane derivative, from the blood [5]. This amantadine transport system is different from typical cation transporters, such as the neutral and basic amino acid transporter (ATB^0,+^/Slc6a14), organic cation transporters 1-2 (OCT1-2/slc22a1-2), multidrug and toxin extrusion protein 1 (MATE1/Slc47a1), carnitine/organic cation transporters 1-2 (OCTN1-2/slc22a4-5), and plasma membrane monoamine transporter (PMAT/slc29a4). Moreover, based on the analysis of kinetic inhibition, pH-dependence, and L-carnitine sensitivity, the amantadine transport system is likely to be different from other putative cation transport systems of the inner BRB, which transport verapamil, clonidine, and propranolol [3,4,5]. Adamantane derivatives, such as amantadine and memantine, are known to exert neuroprotective effects by inhibiting N-methyl-D-aspartate (NMDA) receptors [6,7,8,9] and are used for the treatment of neurodegenerative disorders of the brain [10,11,12,13]. Previous studies showed that the overactivation of NMDA receptors caused damage to retinal ganglion cells [14,15,16] and it is considered a mechanism of retinal neurodegeneration under pathological conditions, such as glaucoma [17,18,19]. Moreover, in vivo and in vitro analyses demonstrated that memantine exerted neuroprotective effects against retinal diseases [20,21]. These pieces of evidence indicate that amantadine analogs are promising as retinal neuroprotective agents for the treatment of retinal diseases. In order to achieve effective pharmacotherapy results with the peripheral administration of neuroprotective agents, the efficient retinal delivery of the drugs is necessary. Since the amantadine transport system at the inner BRB is suggested to contribute to the facilitative retinal distribution of its substrates, a more thorough understanding of the compound recognition of this transport system will be helpful to achieve efficient neuroprotectants delivery to the retina and improve retinal disease pharmacotherapies.

The objective of this study was to characterize the detailed compound recognition properties of amantadine-sensitive drug transport systems for efficient retinal drug delivery and treatment of retinal diseases. To investigate the structural features of compounds that are required for the interaction with the amantadine transport system at the inner BRB, we performed an inhibition analysis of [^3^H]amantadine uptake by inner BRB model cells (TR-iBRB2 cells) [22] using aliphatic amine, including adamantane derivatives. In addition, the kinetic inhibition analysis of amantadine uptake was performed to examine the manner of interaction between the amantadine transport system and inhibitors, which present strong inhibitory effects on amantadine uptake occurring at the inner BRB.

## 2. Results

### 2.1. Inhibitory Effects of Aliphatic Amines on Amantadine Uptake by TR-iBRB2 Cells

Structures and physiochemical properties of test compounds are summarized in Figure 1 and Table 1, respectively. Using TR-iBRB2 cells, the inhibitory effect on the uptake of [^3^H]amantadine was performed, and the results are presented in Figure 2 and Table 1. The inhibition ratio of propranolol uptake at the inner BRB has been reported to be partially correlated with the lipophilicity of inhibitors [4]. Similarly, there was a moderate correlation between the lipophilicity (log P) of test compounds and their inhibitory effect on [^3^H]amantadine uptake (Figure 2, correlation coefficient r = −0.689). In the presence of several adamantane amines, such as compounds **1**–**3**, **8**, and **9**, at a concentration of 200 µM, [^3^H]amantadine uptake was significantly inhibited by 71–95%, which was greater than the inhibitory effect of 200 µM unlabeled amantadine (61%). On the other hand, several adamantane derivatives, which have an imidazole group (compound **4**), hydroxy group (compounds **5**,**7** and **10**), two amino groups (compound **6**), or no polar functional group (compounds **25** and **26**) on the adamantane skeleton, presented little inhibitory effect (<10%). Cyclic and linear alkylamines, such as compounds **11**–**17** and **19**–**23**, significantly reduced [^3^H]amantadine uptake by 31–88% at 200 µM, whereas compound **18**, which has an amino and carboxy group, had little effect. Compound **24** (nicotine), which is a tertiary amine and reported to actively transfer to the retina via the inner BRB (K_m_ value in TR-iBRB2 cells = 492 µM) [23], exhibited a moderate inhibitory effect (41%) at a concentration of 1 mM.

Although compounds **5**, **7**, **10**, **25**, and **26** showed over 100% amantadine uptake, the change of the values was not statistically significant compared with the control. Moreover, the maximum value obtained was 116%, which is comparable to the standard deviation range of the control (100 ± 16%). Therefore, it is implied that these compounds do not interact with the amantadine transport system at the inner BRB.

### 2.2. Effects of Compounds ***9***, ***14***, and ***23*** on Amantadine Uptake by TR-iBRB2 Cells

Since these compounds showed the strongest inhibitory effects on [^3^H]amantadine uptake among the groups with the same lipophilic carbon structure, we determined the concentration-dependent inhibitory effects of compounds **9** (adamantane amine), **14** (cycloaliphatic amine), and **23** (linear aliphatic amine) on the amantadine uptake. All compounds showed a concentration-dependent inhibitory effect on [^3^H]amantadine uptake, with IC_50_ values of 22.6 ± 8.9 µM for compound **9**, 67.0 ± 23.5 µM for compound **14**, and 55.3 ± 18.9 µM for compound **23** (Figure 3 and Table 2). Amantadine uptake by TR-iBRB2 cells without inhibitors (control) was composed of saturable and non-saturable components, with a *K*_m_ value of 40.6 ± 11.5 µM, *V*_max_ value of 1.04 ± 0.21 nmol/(min·mg protein), and *K*_d_ value of 2.73 ± 0.37 µL/(min·mg protein) (Figure 4 and Table 2). Compound **9**, at 25.0 µM, and compound **23**, at 62.5 µM, significantly increased the *K*_m_ values to 137 ± 35 µM and 135 ± 34 µM, respectively, without presenting significant changes in the *V*_max_ and *K*_d_ values (Figure 4A,C, and Table 2). These results indicate that compounds **9** and **23** competitively inhibited amantadine uptake with a *K*_i_ value of 7.19 ± 0.61 µM and 27.3 ± 2.6 µM, respectively. In contrast, compound **14**, at the concentration of 55 µM, significantly increased the *K*_d_ value; however, it had no significant effect on other kinetic parameters, suggesting no competitive inhibition of amantadine uptake (Figure 4B and Table 2).

## 3. Discussion

In this study, the structural requirements for compound recognition by the amantadine transport system were investigated. Amantadine is composed of a cationic amino group and a lipophilic adamantane skeleton. To investigate the importance of the cationic amino group, ten adamantane derivatives with modifications around the amino group, or with other substitution groups on the adamantane skeleton, were used. In addition, sixteen other aliphatic amines with a cycloalkane or linear alkyl chain were used to characterize the structural features of the lipophilic moiety required for the interaction with the amantadine transport system. Our results revealed that the amantadine transport system favors amines, which show high lipophilicity and have no polar functional moieties, such as hydroxy and carboxy groups, other than an amino group on the adamantane skeleton or hydrocarbon ring. In addition, in vitro kinetic analyses showed that certain types of compounds, which have an adamantane skeleton or linear alkyl chain, competitively inhibit amantadine uptake with higher affinity compared to amantadine, suggesting that these compounds can be better substrates for the amantadine transport system.

Under physiological conditions, amantadine exists mostly in the cationic form [24]. In addition, our previous results indicated that the amantadine transport system is sensitive to cationic compounds [5]. The present inhibition study (Figure 2 and Table 1) also showed that several cationic adamantane derivatives that have an amino group on the adamantane skeleton (compounds **1**–**3** and **8**) inhibited [^3^H]amantadine uptake by TR-iBRB2 cells. On the other hand, compounds without a cationic moiety on the adamantane skeleton, such as compounds **25** and **26**, showed little effect on [^3^H]amantadine uptake, suggesting the importance of the cationic group for compound recognition of the amantadine transport system. Meanwhile, the compound with a cationic imidazole moiety between the two adamantane skeletons (compound **4**) had no effect on the uptake. Thus, it is implied that the amantadine transport system does not interact with the imidazole moiety or the permanent cation (quaternary ammonium moiety) of the compound. Secondary and tertiary amines (compounds **19** and **20**) had smaller inhibitory effects on [^3^H]amantadine uptake than primary amines, such as compound **14**. This result suggests that the amantadine transport system is more sensitive to primary amines than secondary and tertiary amines. This result was supported by the weak inhibitory effect of compound **24** (nicotine), which is a tertiary amine.

The present study suggests that lipophilic hydrocarbon structures other than adamantane can interact with the amantadine transport system at the inner BRB, since cycloaliphatic and linear aliphatic amines (compounds **11**–**17** and **19**–**23**) significantly reduced [^3^H]amantadine uptake. These inhibitory effects were greater with a higher number of carbons and higher lipophilicity (Figure 2 and Table 1). Although the mechanisms of action of this transport system are not entirely clear, it is implied that compound lipophilicity is one of the key factors for interacting with the amantadine transport system. Compound **14**, which has a cyclohexane moiety with a lower number of carbon and lipophilicity than compounds **15** and **16**, showed the strongest inhibitory effect among the cycloaliphatic amines (compounds **11**–**16**), along with compound **16**. A previous study reported that cyclohexylamine and its derivatives are partially similar in conformation to amantadine and can interact with NMDA receptors [25]. This implies that the conformation of the lipophilic hydrocarbon structure is important to interact with the amantadine transport system as well as NMDA receptors.

Adamantane amines, which have a hydroxy group on the adamantane skeleton (compounds **5**, **7**, and **10**), exhibited no inhibitory effect (Figure 2 and Table 1). In addition, cyclohexylamine, which has a carboxy group on position 1 of the cyclohexane structure (compound **18**), showed little effect. These results suggest that non-amine polar moieties restrict the interaction of compounds with the amantadine transport system. Interestingly, the compound with 2 amino groups in positions 1 and 3 of the adamantane skeleton (compound **6**) had little inhibitory effect on amantadine uptake, suggesting that the number of amino groups on the adamantane skeleton is critical for compound recognition exhibited by the amantadine transport system. Among the adamantane derivatives, compound **9** presented the strongest inhibitory effect, followed by compounds **2**, **3**, and the others. Compound **9** has three atoms between the amino group and the adamantane skeleton, while compounds **2** and **3** have one atom, and the others have an amino group directly on the adamantane skeleton. These results suggest that the distance between the amino group and the adamantane skeleton can affect the interaction of compounds with the transport system.

The kinetic inhibition analysis of amantadine uptake was performed using compounds **9**, **14**, and **23**, which have different hydrophilic carbon structures and present strong inhibitory effects (Table 1). Although compound **3** showed the second strongest inhibitory effect of the test compounds, to clarify the structural characteristics of the potential substrate for the amantadine transport system we determined that analysis with compounds having different hydrophilic carbon structures (adamantane, cycloalkane, and alkyl chains) would be a high priority. Thus, compounds **14** (cycloaliphatic amine) and **23** (linear aliphatic amine), but not compound **3**, were used. The IC_50_ values of compounds **9**, **14**, and **23** for amantadine uptake were calculated to be 22.6, 67.0, and 55.3 µM, respectively (Figure 3 and Table 2), and these values were smaller than the *K*_m_ value (70.9 µM) of amantadine [5]. This result suggests that these compounds interact with the amantadine transport system with higher affinity than amantadine. In addition, compounds **9** and **23** presented competitive inhibition of amantadine uptake, with *K*_i_ values of 7.19 and 27.3 µM, respectively (Figure 4 and Table 2), suggesting that compounds **9** and **23** are substrate candidates for the amantadine-sensitive drug transport system at the inner BRB. In our previous study, verapamil, which is actively distributed to the retina via the inner BRB, showed strong inhibition of amantadine uptake; however, it was not competitive inhibition [5]. Thus, these results imply that compounds **9** and **23** have better potential as substrates than verapamil for this drug transport system. In contrast, compound **14** did not inhibit the amantadine uptake competitively (Figure 4 and Table 2). Although the detailed mechanisms of the interaction between compound **14** and the amantadine transport system remain unclear in the literature, future studies will clarify whether cyclohexane amines are transported via the amantadine transport system at the inner BRB. In summary, these results suggest that the amantadine-sensitive drug transport system at the inner BRB closely interacts with lipophilic primary amines with an adamantane skeleton, hydrocarbon ring, or linear alkyl chain (Figure 5). In addition, adamantane amines and linear alkyl amines have great potential for active transfer to the retina via the inner BRB utilizing this transport system. In particular, compound **9** showed the highest inhibitory effect and affinity to the amantadine transport system of the test compounds. Therefore, further studies, including a detailed transport analysis of compound **9** and related compounds, could lead to a better understanding of retinal drug delivery via the amantadine transport system.

Considering the potential pharmacological use of the compounds/drugs, the effect of the amantadine-sensitive drug transport system on distribution and excretion in tissues other than the retina is important. Amantadine has been reported to be distributed to the brain and almost all is excreted by the kidneys. Although the transport mechanisms of amantadine to these tissues are not entirely clear, ATB^0,+^, MATE1, and OCT1-2 are suggested to be involved [26,27,28]. However, at the inner BRB, the amantadine transport system has different transport characteristics from these well characterized transporters [5]. Thus, the amantadine-sensitive drug transport system is useful for retinal-targeted drug delivery, and their candidate substrates could be promising therapeutic agents for retinal diseases.

A previous study reported that amantadine and memantine (compound **8**) bind to the phencyclidine (PCP) binding site of the NMDA receptor and exert pharmacological effects [29,30,31]. In addition, pharmacophore and quantitative structure–affinity relationship analyses showed the structural requirement for binding to the PCP binding site: an amine moiety with an aromatic ring or aliphatic hydrocarbon structure, such as an adamantane and cycloalkane skeleton [25,32,33]. These structural features are similar to those of the test compounds, which exhibited strong inhibitory effects on the amantadine transport system in this study (Figure 1 and Table 2). Although the present study did not determine whether test compounds inhibit NMDA receptors, the similarity in the structural requirement for the compound–amantadine transport system and compound–NMDA receptor interactions suggests that test compounds might be able to inhibit NMDA receptors. Our results could be helpful in synthesizing NMDA receptor inhibitors that are actively transferred to the retina from the blood via the inner BRB. Since these NMDA receptor inhibitors exhibit both high permeability to the retina and neuroprotective effects, they are expected to serve as potential drug candidates for retinal neurodegenerative diseases such as glaucoma. In addition to neuroprotectants, these structural features required for compound recognition by the amantadine transport system will also lead to the development of a variety of drugs that are more actively distributed to the retina using peripheral administration.

## 4. Materials and Methods

### 4.1. Reagents

[^3^H]Amantadine (0.2 Ci/mmol) was purchased from Moravek (Brea, CA, USA). The test compounds that were used to perform the inhibition analysis were obtained from Tokyo Chemical Industry (Tokyo, Japan). Other compounds were commercially available.

### 4.2. Cell Culture and Uptake Analysis

We established TR-iBRB2 cells [22] and regularly utilized the cell line to investigate drug transport mechanisms via the inner BRB [3,4,5,23]. The cells (passage number, 42–49) were cultured using Dulbecco’s modified Eagle’s medium with 10% fetal bovine serum, 20 mM NaHCO_3_, 0.14 mM streptomycin sulfate, and 0.19 mM benzylpenicillin potassium under 5% CO_2_/Air at 33 °C. Uptake studies were performed as previously reported [5]. Briefly, the cells were seeded onto 24-well plates coated with collagen I (BioCoat^TM^ Collagen I Cellware, Corning, NY, USA) at a density of 1.0 × 10^5^ cells/well, and were further incubated for 2 days under the above conditions before uptake analysis. Using extracellular fluid (ECF) buffer (NaCl 122 mM, NaHCO_3_ 25 mM, KCl 3 mM, CaCl_2_ 1.4 mM, MgSO_4_ 1.2 mM, K_2_HPO_4_ 0.4 mM, D-glucose 10 mM, and 4-(2-hydroxyethyl)-1-piperazineethanesulfonic acid 10 mM) at 37 °C and pH 7.4, cells were rinsed three times. The uptake reaction was started by adding 0.1 µCi/well [^3^H]amantadine in the ECF buffer with or without test compounds at 37 °C. Since test compounds **25** and **26** had low water solubility, these compounds were dissolved in 100% DMSO and diluted in ECF buffer for use in the uptake analysis (final concentration: 200 µM, 1% DMSO). This reaction was terminated by washing the cells with ice-cold ECF buffer three times. After washing, the cells were lysed with NaOH (1N) and then neutralized with HCl (1N).

Measurements of [^3^H]amantadine-derived radioactivities were performed using a liquid scintillation counter (LSC-6100, Aloka, Tokyo, Japan). Using a DC protein assay kit (Bio-Rad; Hercules, CA, USA), the cellular protein content was measured. Uptake activities of [^3^H]amantadine were evaluated using the cell/medium ratio from Equation (1), and the results are indicated as a percentage of the control.
Cell/medium ratio = ([^3^H]amantadine (dpm per mg protein) in the cell)/([^3^H]amantadine (dpm per µL) in the medium)(1)

The data analysis of kinetics was performed using MULTI, a non-linear least-squares regression analysis program [34]. The half-maximal inhibitory concentration (IC_50_) was calculated using Equation (2). *P*_mim_, [*I*], and *n* are the inhibitor insensitive components of the relative [^3^H]amantadine uptake with inhibitors, inhibitor concentration, and Hill constant, respectively. *P* and *P*_max_ are the relative [^3^H]amantadine uptake values with or without inhibitors, respectively.
*P* = (*P*_max_ − *P*_min_)/[(1 + ([*I*]/IC_50_)*^n^*] + *P*_min_(2)

Kinetic parameters, including the maximal uptake rate (*V*_max_), Michaelis–Menten constant (*K*_m_), and non-saturable uptake clearance (*K*_d_), were estimated using MULTI based on Equation (3). *V* and *S* are the uptake rate of amantadine and total amantadine concentration, respectively.
*V* = *V*_max_ × *S*/(*K*_m_ + *S*) + *K*_d_ × *S*(3)

The inhibition constant (*K*_i_) of the test compounds, which exhibited the competitive inhibition of the amantadine uptake, was determined using Equation (4):*V* = *V*_max_ × *S*/(*K*_m_ × (1 + [*I*]/*K*_i_) + *S*) + *K*_d_ × *S*(4)

## 5. Conclusions

In the present study, we characterized the structural features of compounds, which are recognized by the amantadine transport system at the inner BRB, and determined that the amantadine transport system is sensitive to lipophilic primary amines possessing an adamantane skeleton, hydrocarbon ring, or linear alkyl chain. In addition, both lipophilic adamantane amines and linear aliphatic amines are suggested to be potential substrates in the amantadine transport system with a higher affinity than amantadine. These results will help scholars to better understand cationic drug transport activity across the inner BRB and improve the delivery system of cationic neuroprotectants, such as NMDA receptor inhibitors, for the treatment of retinal diseases. In future studies, this transport system is expected to be utilized for drug delivery by elucidating the molecular entities and optimizing the substrate compounds.

## Figures and Tables

**Figure 1 pharmaceuticals-16-00435-f001:**
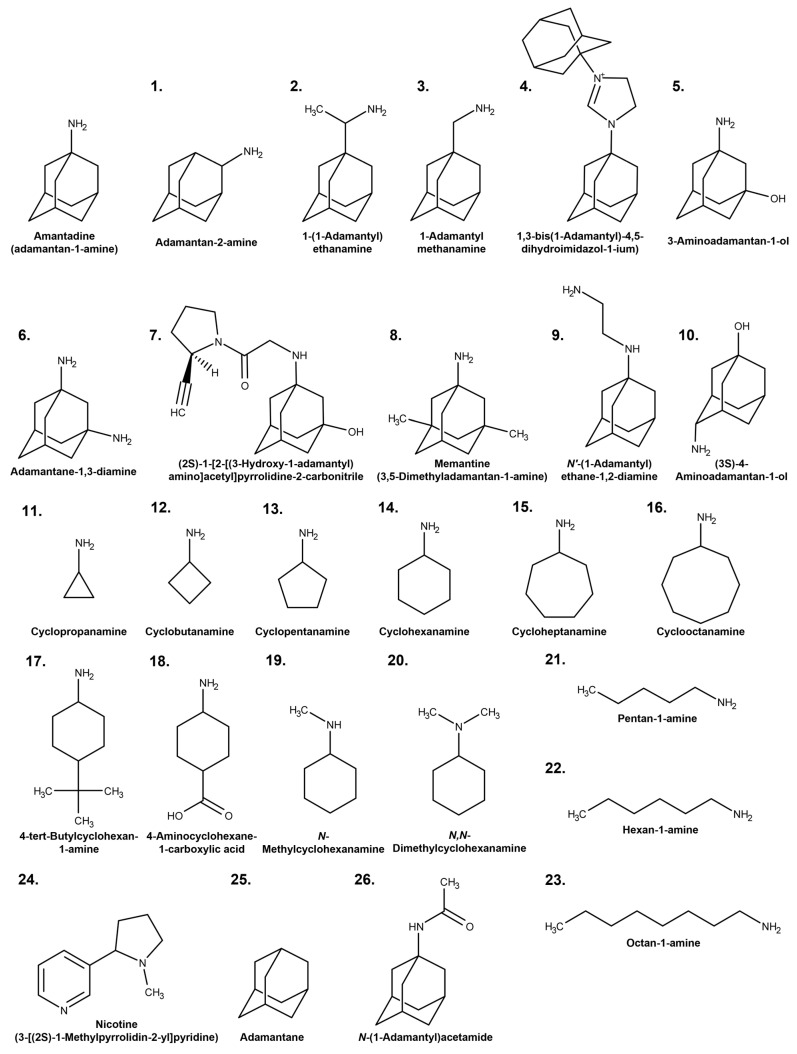
Structures of test compounds. The names of the compounds are according to IUPAC nomenclature. The compound numbers in this figure correspond to the test compounds in Table 1.

**Figure 2 pharmaceuticals-16-00435-f002:**
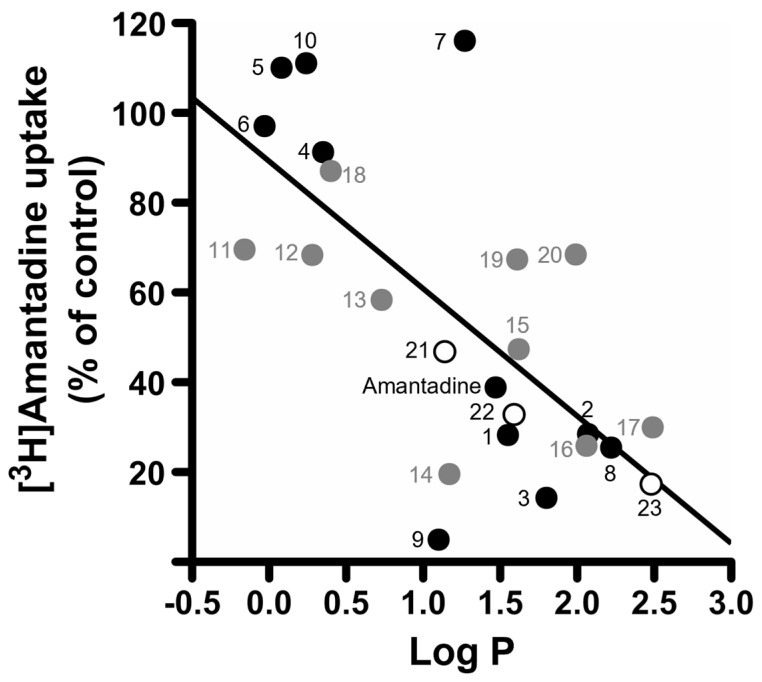
Comparison of the inhibitory effects of test compounds on [^3^H]amantadine uptake and their lipophilicity (log P). Compounds are classified into adamantane derivatives (amantadine and **1**–**10**, closed circles), cycloaliphatic amines (**11**–**20**, gray circles), and linear aliphatic amines (**21**–**23**, open circles). [^3^H]Amantadine uptake by TR-iBRB2 cells and log *p* values are presented in Table 1. The solid line indicates the regression line ([^3^H]amantadine uptake (%) = −28.4 × log P + 89.3, the correlation coefficient r = −0.689).

**Figure 3 pharmaceuticals-16-00435-f003:**
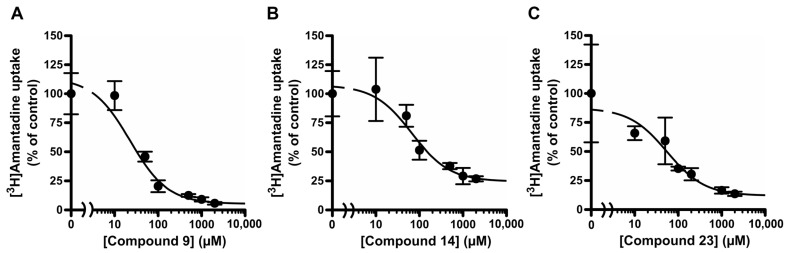
Inhibitory effect of test compounds **9**, **14**, and **23** on [^3^H]amantadine uptake by TR-iBRB2 cells. The uptake of [^3^H]amantadine was examined for 3 min at 37 °C with 1–2000 µM test compounds **9** (**A**), **14** (**B**), or **23** (**C**). The solid lines represent the line fitted to Equation (2) using MULTI. Each point represents the mean ± S.D. (n = 3).

**Figure 4 pharmaceuticals-16-00435-f004:**
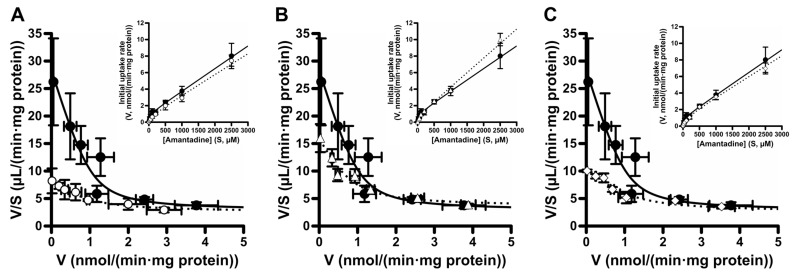
Kinetic inhibition analysis of amantadine uptake into TR-iBRB2 cells with test compounds **9**, **14**, and **23**. Concentration-dependent uptake of amantadine (2–2500 µM) was examined using TR-iBRB2 cells without (control, closed circles) or with compounds **9** ((**A**), 25 µM, open circle), **14** ((**B**), 50 µM, open triangle), and **23** ((**C**), 62.5 µM, open square). The data are represented by Michaelis–Menten (inset) and Eadie–Scatchard plots. The solid and dotted lines in this figure show concentration-dependent amantadine uptake without or with test compounds fitted to Equation (3) using MULTI. Each point represents the mean ± S.D. (n = 3–12).

**Figure 5 pharmaceuticals-16-00435-f005:**
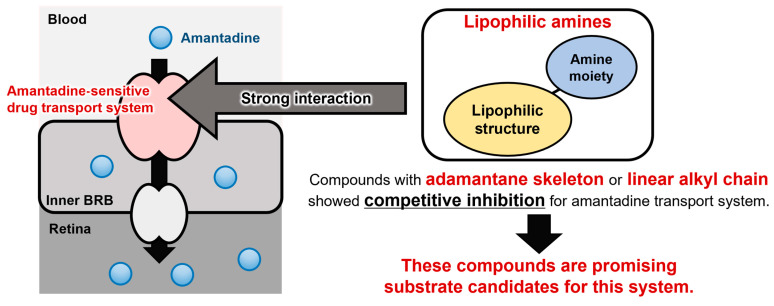
The comprehensive figure of structural features required for interaction with amantadine-sensitive drug transport system at the inner BRB.

**Table 1 pharmaceuticals-16-00435-t001:** Physiochemical properties and inhibitory effects of test compounds on [^3^H]amantadine uptake.

Test Compounds	Formula	MW	pKa	Log P	[^3^H]Amantadine Uptake (% of Control)		
Control					100	±	16
Amantadine	C_10_H_17_N	151.25	10.71	1.47	38.8	±	13.8 **
**1**	C_10_H_17_N	151.25	10.54	1.55	28.2	±	2.1 **
**2**	C_12_H_21_N	179.31	10.14	2.22	25.4	±	0.9 **
**3**	C_11_H_19_N	165.27	9.90	1.80	14.2	±	4.2 **
**4**	C_23_H_35_N_2_	339.55	-	0.35	91.2	±	7.4
**5**	C_10_H_17_NO	167.25	10.34	0.08	110	±	11
**6**	C_10_H_18_N_2_	166.27	8.92, 10.57	−0.03	97.0	±	31.6
**7**	C_17_H_25_N_3_O_2_	302.42	8.79	1.27	116	±	35
**8**	C_12_H_20_O	179.31	10.70	2.07	28.4	±	2.8 **
**9**	C_12_H_22_N_2_	194.32	7.27, 10.53	1.10	4.89	±	1.28 **
**10**	C_10_H_17_NO	167.25	10.23	0.24	111	±	13
**11**	C_3_H_7_N	57.10	9.36	−0.16	69.5	±	22.6 **
**12**	C_4_H_9_N	71.12	10.25	0.28	68.3	±	3.3 **
**13**	C_5_H_11_N	85.15	10.45	0.73	58.3	±	14.1 **
**14**	C_6_H_13_N	99.18	10.45	1.17	19.5	±	2.2 **
**15**	C_7_H_15_N	113.20	10.45	1.62	47.3	±	20.9 **
**16**	C_8_H_17_N	127.23	10.45	2.06	25.8	±	5.7 **
**17**	C_10_H_21_N	155.29	10.45	2.49	29.9	±	15.4 **
**18**	C_7_H_13_NO_2_	143.19	10.46	0.40	87.0	±	2.9
**19**	C_7_H_15_N	113.20	10.70	1.61	67.3	±	13.5 **
**20**	C_8_H_17_N	127.23	10.22	1.99	68.4	±	10.6 **
**21**	C_5_H_13_N	87.17	10.21	1.14	46.8	±	7.5 **
**22**	C_6_H_15_N	101.19	10.21	1.59	32.8	±	12.5 **
**23**	C_8_H_19_N	129.25	10.21	2.48	17.3	±	3.0 **
**24**	C_10_H_14_N_2_	162.24	8.58	1.16	58.7	±	9.7 **
Control (1%DMSO)					100	±	11
**25**	C_10_H_16_	136.24	-	2.89	106	±	3
**26**	C_12_H_19_NO	193.28	-	1.28	98.1	±	13.2

pKa and log P values were calculated using Marvin sketch 22.18 (ChemAxon, https://www.chemaxon.com. Accessed on 11 October 2022). The uptake of [^3^H]amantadine (2.25 µM, 0.1 µCi/well) by TR-iBRB2 cells was performed at 37 °C for 3 min in the presence or absence (control) of test compounds. The concentrations of compound **24** and other test compounds were 1000 and 200 µM, respectively. Compounds **25** and **26** were solubilized in the 100% DMSO and then diluted in uptake buffer (final concentration: 200 µM, 1% DMSO). Each value represents the mean ± S.D. (n = 3–15). ** indicates significant difference from the control (*p* < 0.01).

**Table 2 pharmaceuticals-16-00435-t002:** Kinetic parameters of amantadine uptake by TR-iBRB2 cells with test compounds.

ID	IC_50_ (µM)			*V*_max_ (nmol/(min·mg Protein))			*K*_m app_ (µM)			*K*_d_ (µL/(min·mg Protein))			*K*_i_ (µM)		
Control				1.04	±	0.21	40.6	±	11.5	2.73	±	0.37			
**9**	22.6	±	8.9	0.795	±	0.187	137	±	35 **	2.51	±	0.20	7.19	±	0.61
**14**	67.0	±	23.5	0.787	±	0.171	63.2	±	17.4	3.50	±	0.29 **			
**23**	55.3	±	18.9	1.05	±	0.24	135	±	34 **	2.50	±	0.24	27.3	±	2.6

Each value represents the mean ± S.D. (n = 8). ** indicates significant difference from the control (*p* < 0.01).

## Data Availability

The data of this study are available from the corresponding author upon reasonable request.

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
