# Peer review of "The Structural Characteristics of Compounds Interacting with the Amantadine-Sensitive Drug Transport System at the Inner Blood–Retinal Barrier"

_pharmaceuticals, 2023, doi:10.3390/ph16030435_

Round 1

Reviewer 1 Report

1. Prepare a list of all abbreviations and put them at the beginning of the article.

2. A comprehensive figure of all the mechanisms and effects mentioned in the article should be included in the article.

3. In the results section, it is necessary to use more up-to-date scientific sources.

Author Response

 We would like to thank four reviewers for providing constructive comments regarding the improvement of the original manuscript. Regarding the concerns from the reviewers, we have prepared the point-by-point responses to the reviewer’s comments. The revised points in the manuscript were marked up using the “TrackChanges” function of MS Word.

  1. Prepare a list of all abbreviations and put them at the beginning of the article.

 Following your kind comment, we prepare the list of all abbreviations before the Introduction section.

  1. A comprehensive figure of all the mechanisms and effects mentioned in the article should be included in the article.

 Thank you for your suggestion. To help readers understand the conclusions of this study, we prepared comprehensive figure in the discussion section. We are grateful if you satisfy this Figure.

  1. In the results section, it is necessary to use more up-to-date scientific sources.

 Thank you for your critical comment. In the result section, we cited the reference about the relationship between the inhibition rate of cationic compound uptake and lipophilicity of inhibitors by Kubo et al. (J. Pharm. Sci. 2013, 102, 3332-3342). With regard to analyses of relationship between the compound lipophilicity and inhibition of drug transport in the inner BRB model cells, this reference is the latest one among manuscripts. Thus, we have decided this reference for the use in the Result section. In addition, we have added updated references throughout this paper considering the reviewer’s helpful comment.

Finally, thank you very much again for your valuable and helpful comments.

Reviewer 2 Report

The manuscript prepared by Ken-ichi Hosoya and co-workers describes the structural features of compounds for the amantadine-sensitive transport system. Authors studied the inhibitory effect of 26 compounds (25 amines and adamantane, respectively) on amantadine uptake by TR-iBRB2 cells. Moreover, three compounds were further investigated in more detail. The topic of this manuscript could be important and of interest to researchers dealing with the medicinal chemistry of adamantane-based derivatives. In my opinion, the manuscript is relatively brief and should be published as a short communication rather than an article.

Despite all of comments mentioned in my review (see the attached file), I think that manuscript could be, after its revision, accepted for the publication in Pharmaceuticals journal.

Author Response

 We would like to thank four reviewers for providing constructive comments regarding the improvement of the original manuscript. Regarding the concerns from the reviewers, we have prepared the point-by-point responses to the reviewer’s comments. The revised points in the manuscript were marked up using the “TrackChanges” function of MS Word.

Response to Reviewer #2

1) Table 1:

  • I suggest to change Table 1 to the Figure, because the Table takes up too much space (3 pages)
  • The size and drawing of presented structures should be unified (e.g. compounds 4, 7, 17, 25, 26, …)
  • The nomenclature of the presented compounds should be improved (according to IUPAC recommendations) and unified; e.g compd. 5 should be named as 3-aminoadamantan-1ol, compd. 7 (change the systematic name for “adamantyl”), compd. 11–17 (it should be cycloalkanamines, not cycloalkylamines, e.g. cyclopropanamine for compd. 11, and so on) compd. 18 (why it is listed as “cis” derivative?), compd. 19–23 (it should be alkanamines, not alkylamines, e.g. N-methylcyclohexanamine for compd. 19, and so on), compd. 24 (the systematic name should be also listed)

 Thank you for your kind comment. We have changed Table 1 to Figure 1 with revising the skeletal formula and name of the test compounds. We are grateful if you satisfy the revision of this Figure 1.

2) Compound 24 was tested at the concentration of 1 mM (line 93). The rest of studied compounds were tested at the concentration of 200 µM. What was the reason for the different concentrations used? Were the studied compounds tested at both mentioned concentrations or not? If so, can you show the results for both concentrations used?

 It has been reported that compound 24 (nicotine), which is actively transferred to the retina via inner BRB, is taken up into TR-iBRB2 cells with a Km value of 492 µM1. Based on the evidence, the concentration of compound 24 in this study was determined as 1 mM to examine the inhibition degree to the amantadine uptake by TR-iBRB2 cells.

 At the concentration of 200 µM, unlabeled amantadine significantly reduced [3H]amantadine uptake by TR-IBRB2 cells. To compare the inhibitory effect of unlabeled amantadine, other test compounds were used at the concentration of 200 µM. Following your valuable comment, we have added the information to the results section (p. 3, lines 102-105).

Revised manuscript (Page 3, Line 102-105; Results)

 Compound 24 (nicotine), which is a tertiary amine and reported to actively transfer to the retina via the inner BRB (Km value in TR-iBRB2 cells = 492 µM) [23], exhibit a moderate inhibitory effect (41%) at the concentration of 1 mM.

Previously-submitted manuscript

 Compound 24 (nicotine), which is a tertiary amine exhibit a moderate inhibitory effect (41%) at the concentration of 1 mM.

3) The statement that the “amantadine transport system does not interact with the imidazole moiety” (line 170) seems relatively strong, because the authors used only one imidazole derivative in this study. Moreover, not only the imidazole moiety, but also the presence of a permanent cation may play an important role in this effect.

 Thank you for giving us the valuable comment. To indicate the importance of amine moiety for interacting with the amantadine transport system, this statement was used in this part. Considering the permanent cation of imidazole and other adamantanes with imidazole moiety, it would be more accurate for the reader to change the wording and add the explanation. We have revised this part (p. 8, lines 187-189).

Revised manuscript (Page 8, Line 187-189; Discussion)

 Meanwhile, the compound with a cationic imidazole moiety between the two adamantane skeletons (compound 4) had no effect on the uptake. Thus, it is implied that the amantadine transport system does not interact with the imidazole moiety or the permanent cation (quaternary ammonium moiety) of the compound.

Previously-submitted manuscript

 On the other hand, the compound with a cationic imidazole moiety between the two adamantane skeletons (compound 4) had no effect on the uptake, suggesting that the amantadine transport system does not interact with the imidazole moiety of the compound.

4) The authors state that the compound 14 showed the strongest activity among cycloaliphatic amines (lines 182–183). I think that compound 16 shows the same inhibitory activity as compound 14.

 As the reviewer pointed out, compounds 14 and 16 strongly reduced [3H]amantadine uptake to 19.5% and 24.8%, respectively. There is no great difference between the inhibitory effect of compounds 14 and 16, however, compound 14 has a lower carbon number and lipophilicity than compound 16. Considering these physiological properties between compounds 14 and 16, compound 14 is considered to be suitable for the following studies. We have added text in lines 201-203 (p. 8) to clearly convey our intent to the reader.

Revised manuscript (Page 8, Line 201-203; Discussion)

 Compound 14, which has a cyclohexane moiety with the lower number of carbon and lipophilicity than compounds 15 and 16, showed the strongest inhibitory effect among the cycloaliphatic amines (compounds 11–16) as well as compound 16.

Previously-submitted manuscript

 Compound 14, which has a cyclohexane moiety, showed the strongest inhibitory effect among the cycloaliphatic amines (compounds 11–16).

5) Can authors explain what this means, if the amantadine uptake is higher than 100%, as seen in Table 2 for compounds 5, 7, 10, 25 and 26?

 Thank you for your comment. The percentage of control is shown as more than 100% if [3H]amantadine uptake is promoted. Although compounds 5, 7, 10, 25, and 26 showed over 100% amantadine uptake, change of the values was not statistically significant compared with control. Moreover, the maximum of the value was obtained to be 116%, which is comparable to the standard deviation range of the control (100 ± 16). Therefore, it is implied that these compounds do not interact with the amantadine transport system at the inner BRB.

 In the revised manuscript, the above points have been partially added for the readers to understand our consideration.

Added sentence (Page 3, Line 106-111; Results)

 Although compounds 5, 7, 10, 25, and 26 showed over 100% amantadine uptake, change of the values was not statistically significant compared with control. Moreover, the maximum of the value was obtained to be 116%, which is comparable to the standard deviation range of the control (100 ± 16). Therefore, it is implied that these compounds do not interact with the amantadine transport system at the inner BRB.

6) Why did the authors not use for the further study also compound 3, which showed the second strongest inhibitory effect (Table 2)?

 Thank you for your helpful comment. As you pointed out, compound 3 has the second strongest inhibitory effect. However, in order to clarify the structural characteristics of the potential substrate for the amantadine transport system, we determined that analysis with compounds having different hydrophilic carbon structures (adamantane, cycloalkane, and alkyl chains) would be a high priority. Thus, we selected the compound with the highest inhibitory effect, but not compound 3. To make this point clearer, we have added text in the discussion section (p. 9, lines 225-230).

Added sentence (Page 9, Line 225-230; Discussion)

 Although compound 3 showed the second strongest inhibitory effect of the test compounds, to clarify the structural characteristics of the potential substrate for the amantadine transport system, we determined that analysis with compounds having different hydrophilic carbon structures (adamantane, cycloalkane, and alkyl chains) would be a high priority. Thus, compounds 14 (cycloaliphatic amine) and 23 (linear aliphatic amine), but not compound 3 were used.

7) The most promising results seem to have been obtained for compound 9. I wonder why the authors did not use other structurally related derivatives, such as N-(1-adamantylpropane-1,3-diamine, 1-adamantylethanamine or 2-adamantylethanamine? I think that further study on the effects of these (and other related) compounds could yield interesting results.

 Thank you for your beneficial recommendation. In this study, we used amine compounds that are widely available and have a basic structure. As a result, compound 9 showed the highest potential as a compound that can be easily transferred to the retina. Therefore, further studies, including the detailed transport analyses of compound 9 and related compounds, could lead to a better understanding of retinal drug delivery via the amantadine transport system. We have added the sentences about the further step to the discussion section (p. 9, lines 250-253). We are grateful if you also satisfy this revised version.

Added sentence (Page 9, Lines 250-253; Discussion)

 Particularly, compound 9 showed the highest inhibitory effect and affinity to the amantadine transport system of test compounds. Therefore, further studies, including detailed transport analysis of compound 9 and related compounds, could lead to a better understanding of retinal drug delivery via the amantadine transport system.

Finally, thank you very much again for your valuable and helpful comments.

 References

  1. Tega, Y; Kubo, Y; Yuzurihara, C; Akanuma, S; Hosoya, K. Carrier-Mediated Transport of Nicotine Across the Inner Blood-Retinal Barrier: Involvement of a Novel Organic Cation Transporter Driven by an Outward H(+) Gradient. Pharm. Sci. 2015, 104, 3069-3075.

Reviewer 3 Report

Reviewer comments

In research article The Structural Characteristics of Compounds Interacting with the Amantadine-Sensitive Drug Transport System at the Inner 3 Blood–Retinal Barrier authors represents compounds which interact with amantadine transport system at the inner BRB. These results may be helpful for producing the appropriate drug design to improve the blood-to-retina delivery of neuroprotective drugs. In some parts, author need to better explain a purpose of the study. Further research is necessary considering that these are possible carriers for the treatment of the disease. What do the authors suggest for further research?

Authors need to remark in text the next line:

Line 25 Replace phrase delivery system with transport carriers or system.

Line 68 Should be better explaining the aim of this research. You should put emphasis on new amantadine-sensitive drug transport system for retinal diseases (lipophilic amines, especially primary amines).

Line 119 In addition to the negative control (In the absence of inhibitors (control…) do you have a compound/drug that crosses the retinal barrier well and compare your results with it? Maybe it is pure Amantadine?

Line 176 Author say „The present study suggests that lipophilic hydrocarbon structures other than adamantane can interact with the amantadine transport system at the inner BRB, since cycloaliphatic and linear aliphatic amines (compounds 11–17 and 19–23) significantly reduce amantadine uptake. These inhibitory effects were greater with a higher number of carbons and higher lipophilicity (Figure 1 and Table 2), suggesting that compound lipophilicity is one of the key factors for interacting with the amantadine transport system.”  This is only possible mechanism or authors suggest that is a mechanism of action of amantadine transport system? Please be clear with statement.

In section Materials and Methods, we don’t have mention 1% DMSO. In line 109 you stated that you used DMSO for dissolution. Cell line TR-iBRB2 cells are purchased from where?

Line 145 In all research article, especially Discussion section, authors need to better explain the sentences and to better explain the scientific contribution of the presented drug transport system. Considering the potential pharmacological use of the drug/compounds, it is necessary to investigate the amantadine-sensitive drug transport system in another cell lines, which is different from well-characterized transporters. Was the experiment done in triplicate?

Line 272 In Conclusion section authors need to concretize what is further research.

According to suggestions, I recommend a major revision of the research article.

Author Response

 We would like to thank four reviewers for providing constructive comments regarding the improvement of the original manuscript. Regarding the concerns from the reviewers, we have prepared the point-by-point responses to the reviewer’s comments. The revised points in the manuscript were marked up using the “TrackChanges” function of MS Word.

Line 25 Replace phrase delivery system with transport carriers or system.

 Following your comment, we have replaced “drug delivery” with “drug transport system” in the keyword section.

Line 68 Should be better explaining the aim of this research. You should put emphasis on new amantadine-sensitive drug transport system for retinal diseases (lipophilic amines, especially primary amines).

 Following your valuable comment, we have added a more detailed explanation of the aim in the last paragraph of the introduction section (p. 2, lines 76-78). We are grateful if you satisfy this revision.

Added sentence (Page 2, Lines 76-78; Introduction)

 The objective of this study is to characterize the detailed compound recognition properties of amantadine-sensitive drug transport systems for efficient retinal drug delivery and treatment of retinal diseases.

Line 119 In addition to the negative control (In the absence of inhibitors (control…) do you have a compound/drug that crosses the retinal barrier well and compare your results with it? Maybe it is pure Amantadine?

 Thank you for your helpful comment. We previously investigated the concentration-dependent uptake of amantadine by TR-iBRB2 cells with or without verapamil, which is actively transported to the retina across the BRB1. In that study, amantadine uptake exhibited saturable and non-saturable processes and verapamil did not show competitive inhibition of this amantadine uptake. Although detailed mechanisms of interaction between verapamil and amantadine-sensitive drug transport system is unclear, compounds 9 and 23 showed competitive inhibition of amantadine uptake, implying that these compounds have better potential as substrates than verapamil for this drug transport system. We have added this in the discussion section (p. 9, lines 237-241).

Added sentence (Page 9, Lines 237-241; Introduction)

 In our previous study, verapamil, which is actively distributed to the retina vis the inner BRB, showed strong inhibition of amantadine uptake, however, it was not competitive inhibition [5]. Thus, these results imply that compounds 9 and 23 have better potential as substrates than verapamil for this drug transport system.

Line 176 Author say ”The present study suggests that lipophilic hydrocarbon structures other than adamantane can interact with the amantadine transport system at the inner BRB, since cycloaliphatic and linear aliphatic amines (compounds 11–17 and 19–23) significantly reduce amantadine uptake. These inhibitory effects were greater with a higher number of carbons and higher lipophilicity (Figure 1 and Table 2), suggesting that compound lipophilicity is one of the key factors for interacting with the amantadine transport system.” This is only possible mechanism or authors suggest that is a mechanism of action of amantadine transport system? Please be clear with statement.

 Thank you for your valuable comment. We think compound lipophilicity is important to interact with the amantadine transport system, but the mechanisms of action of this system are not completely clear in this study. We have revised the text in the discussion section (p. 8, line 198-200) to clarify this point.

Revised manuscript (Page 8, Lines 198-200; Discussion)

 These inhibitory effects were greater with a higher number of carbons and higher lipophilicity (Figure 2 and Table 1). Although the mechanisms of action of this transport system are not entirely clear, it is implied that compound lipophilicity is one of the key factors for interacting with the amantadine transport system.

Previously-submitted manuscript

 These inhibitory effects were greater with a higher number of carbons and higher lipophilicity (Figure 2 and Table 1), suggesting that compound lipophilicity is one of the key factors for interacting with the amantadine transport system.

In section Materials and Methods, we don’t have mention 1% DMSO. In line 109 you stated that you used DMSO for dissolution. Cell line TR-iBRB2 cells are purchased from where?

 Thank you for your kind suggestion. Since compounds 25 and 26 have low water solubility, we use DMSO to solubilize in the uptake buffer. We established TR-iBRB2 cells2 and regularly utilize the cell line to investigate drug transport mechanisms across the inner BRB1. We have added text to the Materials and Methods section (p. 10, lines 292-293 and 304-306).

Added sentence (Page 10, Lines 292-293; Discussion)

 We established TR-iBRB2 cells [22] and regularly utilize the cell line to investigate drug transport mechanisms via the inner BRB [3-5, 23].

Added sentence (Page 10, Lines 304-306; Discussion)

 Since test compounds 25 and 26 have low water solubility, these compounds were dis-solved in 100% DMSO and diluted in ECF buffer for use in the uptake analysis (Final concentration: 200 µM, 1% DMSO).

Line 145 In all research article, especially Discussion section, authors need to better explain the sentences and to better explain the scientific contribution of the presented drug transport system. Considering the potential pharmacological use of the drug/compounds, it is necessary to investigate the amantadine-sensitive drug transport system in another cell lines, which is different from well-characterized transporters. Was the experiment done in triplicate?

 Thank you very much for your excellent suggestion. As you point out, it is important that the effect of amantadine sensitive-drug transport system on drug distribution and excretion in tissues other than the retina. Amantadine is reported to be distributed to the brain and mostly excreted by the kidneys. Some well-characterized transporters, such as MATE1 and OCTs, are involved in the transport to these tissues3-5. However, amantadine transport systems at the inner BRB is different from these transporters1. We have added the text in the discussion section (p. 9, lines 254-263) about the comparison with amantadine transport mechanisms in other tissues.

Added sentence (Page 10, Lines 254-263; Discussion)

 Considering the potential pharmacological use of the compounds/drugs, the effect of the amantadine-sensitive drug transport system on distribution and excretion in tissues other than the retina is important. Amantadine has been reported to be distributed to the brain and almost all is excreted by the kidneys. Although the transport mechanisms of amantadine to these tissues are not entirely clear, ATB0,+, MATE1, and OCT1-2 are suggested to be involved [26-28]. However, at the inner BRB, the amantadine transport system has different transport characteristics from these well characterized transporters [5]. Thus, the amantadine-sensitive drug transport system is useful for retinal-targeted drug delivery, and their candidate substrates could be promising therapeutic agents for retinal diseases.

Line 272 In Conclusion section authors need to concretize what is further research.

 Thank you very much for your important comments. To help readers understand the conclusions of this study, we have added supporting figure in the discussion section and an explanation about the further research in the conclusion section (p. 11, lines 335-337).

Added sentence (Page 10, Lines 335-337; Discussion)

 In future studies, it is expected to be utilized this transport system for drug delivery by elucidating the molecular entities and optimizing the substrate compounds.

Finally, thank you very much again for your valuable and helpful comments.

References

  1. Shinozaki, Y.; Akanuma, S.; Mori, Y.; Kubo, Y.; Hosoya, K. Comprehensive Evidence of Carrier-Mediated Distribution of Amantadine to the Retina across the Blood-Retinal Barrier in Rats. Pharmaceutics 2021, 13.
  2. Hosoya, K.; Tomi, M.; Ohtsuki, S.; Takanaga, H.; Ueda, M.; Yanai, N.; Obinata, M.; Terasaki, T. Conditionally immortalized retinal capillary endothelial cell lines (TR-iBRB) expressing differentiated endothelial cell functions derived from a transgenic rat. Eye. Res. 2001, 72, 163-172.
  3. Goralski, K.B.; Lou, G.; Prowse, M.T.; Gorboulev, V.; Volk, C.; Koepsell, H.; Sitar, D.S. The cation transporters rOCT1 and rOCT2 interact with bicarbonate but play only a minor role for amantadine uptake into rat renal proximal tubules. Pharmacol. Exp. Ther. 2002, 303, 959-968.
  4. Kooijmans, S.A.; Senyschyn, D.; Mezhiselvam, M.M.; Morizzi, J.; Charman, S.A.; Weksler, B.; Romero, I.A.; Couraud, P.O.; Nicolazzo, J.A. The involvement of a Na(+)- and Cl(-)-dependent transporter in the brain uptake of amantadine and rimantadine. Pharm. 2012, 9, 883-893.
  5. Muller, F.; Weitz, D.; Derdau, V.; Sandvoss, M.; Mertsch, K.; Konig, J.; Fromm, M.F. Contribution of MATE1 to Renal Secretion of the NMDA Receptor Antagonist Memantine. Pharm. 2017, 14, 2991-2998.